# Molecular Characterization of Wheat Stripe Rust Pathogen (*Puccinia striiformis* f. sp. *tritici*) Collections from Nine Countries

**DOI:** 10.3390/ijms22179457

**Published:** 2021-08-31

**Authors:** Qing Bai, Anmin Wan, Meinan Wang, Deven R. See, Xianming Chen

**Affiliations:** 1Department of Plant Pathology, Washington State University, Pullman, WA 99164-6430, USA; qing.bai@wsu.edu (Q.B.); anminwan@google.com (A.W.); meinan_wang@wsu.edu (M.W.); deven.see@usda.gov (D.R.S.); 2U.S. Department of Agriculture, Agricultural Research Service, Wheat Health, Genetics, and Quality Research Unit, Pullman, WA 99164-6430, USA

**Keywords:** diversity, migration, molecular markers, population genetics, *Puccinia striiformis* f. sp. *tritici*, stripe rust

## Abstract

Stripe rust, caused by *Puccinia striiformis* f. sp. *tritici* (*Pst*), is one of the most important diseases of wheat worldwide. To understand the worldwide distribution of its molecular groups, as well as the diversity, differentiation, and migration of the *Pst* populations, 567 isolates collected from nine countries (China, Pakistan, Italy, Egypt, Ethiopia, Canada, Mexico, Ecuador, and the U.S.) in 2010–2018 were genotyped using 14 codominant simple sequence repeat markers. A total of 433, including 333 new multi-locus genotypes (MLGs), were identified, which were clustered into ten molecular groups (MGs). The MGs and country-wise populations differed in genetic diversity, heterozygosity, and correlation coefficient between the marker and virulence data. Many isolates from different countries, especially the isolates from Mexico, Ecuador, and the U.S., were found to be identical or closely related MLGs, and some of the MGs were present in all countries, indicating *Pst* migrations among different countries. The analysis of molecular variance revealed 78% variation among isolates, 12% variation among countries, and 10% variation within countries. Only low levels of differentiation were found by the pairwise comparisons of country populations. Of the 10 MGs, 5 were found to be involved in sexual and/or somatic recombination. Identical and closely related MLGs identified from different countries indicated international migrations. The study provides information on the distributions of various *Pst* genetic groups in different countries and evidence for the global migrations, which should be useful in understanding the pathogen evolution and in stressing the need for continual monitoring of the disease and pathogen populations at the global scale.

## 1. Introduction

Stripe rust (yellow rust) caused by *Puccinia striiformis* Westend. f. sp. *tritici* Erikss. (*Pst*) is a destructive disease across major wheat growing regions of the world [1,2,3,4,5]. Yield losses caused by stripe rust vary greatly depending upon the susceptibility of cultivars, the time point of initial infection, the rate of disease development, and the duration of the disease, but can reach up to 100% if highly susceptible cultivars are grown under extremely favorable weather conditions for stripe rust development [4,6,7,8]. Due to the capacity for rapid changes in virulent races to circumvent resistance in wheat cultivars and genotypes adapting to different environments, as well as the long-distance dissemination of *Pst* urediniospores via wind dispersal or human activities, stripe rust is an increasing problem threatening global wheat production [4,6,9,10,11,12,13,14,15].

In the past two decades, a series of severe epidemics caused by stripe rust were reported in large-scale regions, including East Asia, Central and West Asia, East and North Africa, Europe, western Australia, and North and South America [4,5,6,7,16,17,18,19,20]. Since 2000, new races, as well as aggressive and high temperature-adapted strains, have spread the major *Pst* populations worldwide and caused severe damage in terms of yield loss and cost of fungicide applications in many countries. For example, severe epidemics occurred throughout the U.S. in 2000, 2001, 2003, 2005, 2010, 2011, 2015, and 2016 [4,5,17], and in Australia in 2003–2006 [5,21]. In 2010, prevalent stripe rust caused economic losses in East Africa [20], Central and West Asia [5,19,20], and North America [5,6,7,22,23]. For example, a devastating stripe rust epidemic occurred on more than 600,000 ha of wheat in Ethiopia, which led to a cost of more than USD 3.2 million in the application of fungicides [20]; similarly in the U.S., wheat stripe rust caused the most widespread epidemic throughout the whole country in recorded history in 2010, and the extremely severe epidemic in the Pacific Northwest in 2011, which resulted in large-scale applications of foliar fungicides [5,6,7]. Since 2011, the invasive races of *Pst*, “Warrior” and “Kranich” largely destroyed the pre-existing NW European populations. Moreover, the “Kranich” race has been demonstrated to have a high sexual reproduction capacity under suitable conditions, leading to potential threats to wheat production worldwide [14,24,25].

To track the migration routes, determine the origin, and detect changes in genetic groups, several studies have been performed on *Pst* at relatively large geographic scales using different molecular markers and virulence data. Some studies have suggested that the Himalayan region or the whole Mediterranean to Asia region may be the putative center of origin of *Pst*, because of its high genotypic diversity, high ability for sexual reproduction, and the strong differentiation from other populations [1,13,26]. The Asian populations, including those in China, Nepal, and Pakistan, have been indicated as the possible sources of the emergence of new, virulent, and aggressive strains due to the high level of recombination, diversity, and ability for sexual reproduction, the latter of which has been reported in China in several studies [27,28,29,30,31]. Other studies have also tried to track the migration of *Pst* populations and the worldwide distribution of different genetic linkages [15,26,32,33]. However, recent molecular studies on the characterization of international *Pst* collections have been conducted mainly on isolates collected before 2010 [13,33], indicating that more recent populations must be characterized to determine whether the international *Pst* population has undergone any major genetic changes since 2010.

In our group, stripe rust samples are routinely collected and received from throughout the U.S., and the *Pst* isolates are characterized for virulence using a set of wheat differentials and for genotypes using simple sequence repeat (SSR) and other markers [4,22,23,34,35,36,37,38,39,40,41,42,43]. These studies identified a large number of virulence races and multi-locus genotypes (MLGs) in the *Pst* populations, as well as clonal production, relatively high diversities in some epidemiological regions and on grass hosts, differentiation and migration among different regions, and long-term dynamics for inferring evolutionary mechanisms. In addition, the roles of sexual reproduction, somatic reproduction, and mutation in *Pst* variation were studied by developing sexual, somatic recombinant, and mutation populations under controlled conditions, characterized by both virulence testing and molecular markers [41,42,43,44,45,46,47,48]. Our laboratory has received stripe rust samples from other countries and characterized them through virulence testing and using molecular markers [26,49,50,51,52]. Previous studies characterized *Pst* isolates collected from 16 countries in 2006–2010 through virulence testing [50] using co-dominant SSR markers [26]. These studies identified large numbers of races [50] and genotypes, and revealed overall high diversity and migrations among different countries [26]. Since 2010, we have received stripe rust samples from eight countries (Canada, China, Ecuador, Egypt, Ethiopia, Italy, Mexico, and Pakistan), in addition to the U.S. collections. The virulence data from the collections from Ethiopia in 2013 and 2014 [51], and those from the collections from other countries collected from 2013 to 2020, except the data from the U.S. collections from 2013 to 2020, have recently been published [52], but the molecular characterization of the foreign collections has not been published. Based on the virulence data of these collections, it was hypothesized that *Pst* migrations have occurred among different countries and continents in the recent years. The present study was conducted to test the hypothesis via the molecular characterization of the collections. 

The specific objectives of this study were to (1) characterize the international collections of *Pst* since 2010 using SSR markers and identify molecular groups, (2) determine and compare the levels of genetic diversities, heterozygosity, and differentiation among different molecular groups and countries, and (3) track migrations among *Pst* populations among different countries. Using the molecular data, we also inferred reproduction mode and the correlation between the molecular data and previously published virulence data [51,52]. The results should provide a better understanding of stripe rust epidemiology and distribution, and the migration of the pathogen populations, as well as mechanisms of pathogen evolution.

## 2. Results

### 2.1. Multi-Locus Genotypes

A total of 433 MLGs were detected in the 567 *Pst* isolates using the 14 SSR loci. The allelic genotypes of the 567 isolates at the 14 marker loci and their assigned MLGs are provided in Appendix A, and those of the 433 MLGs, together with the numbers of isolates and the country distribution, are given in Appendix A. Among the 433 MLGs, 53 were detected in 2 to 29 isolates, and 380 each were detected in only 1 isolate. Only 15 MLGs were detected in two or more countries. Of the 433 MLGs, 100 previously identified MLGs were from the U.S. isolates used in the present study, and 333 MLGs were identified for the first time in isolates of the other countries.

### 2.2. Minimum Spanning Network of the MLGs

A minimum spanning network was constructed to show the country distribution and genetic relatedness among the MLGs based on the Bruvo’s distance values using a stepwise mutation model for the SSR loci (Figure 1). Most MLGs within each country were more closely clustered together, while MLGs from different countries were respectively separated. The 433 MLGs could be separated into six branches, which are labeled by the broken lines in Figure 1. In the first branch (B1), most MLGs from Italy were clustered in the center of the network and were highly connected to other MLGs from the other countries. The large branch (B2) at the bottom of the Figure contains the most diverse MLGs, mainly from China, with some from Italy, the U.S., and Mexico, and few MLGs from the remaining countries, suggesting the wide spreading of this genetic group. The MLGs from the U.S., Mexico, and Ecuador were most closely related in a separate branch (B3), suggesting frequent gene flow in the continents of North and South America, while another closely related branch (B4) mainly contained MLGs from the U.S. and Italy. Most Ethiopian MLGs formed a distinct branch (B5) that was the least related to other countries, except for another small branch (B6) of MLGs from Mexico and a few from Ecuador, Canada, and Italy.

### 2.3. Molecular Groups (MGs)

To further determine the relationships of the collections, a hierarchical cluster analysis of the 567 isolates based on the 14 SSR markers was conducted. These isolates or the 433 MLGs were optimally clustered into ten MGs (*K* = 10). The isolates and MLGs in each MG are provided in Appendix A, respectively. The total number of isolates and the number of isolates in each MG are presented in Table 1. The genetic relationships and distributions in various countries of the MGs are shown in Figure 2. The number of isolates in each MG ranged from 14 (2.5%) in MG4 to 169 (29.8%) in MG9 (Table 1). MG1, MG2, and MG3 were closely related, and they formed a branch distinct from the other MGs (Figure 2). MG1 and MG3 had 58.9% and 50.0% of the isolates, respectively, from Italy, as well as 41.1% and 50.0% of the isolates, respectively, from other countries, except Pakistan. MG2 had 83.3% of the isolates from Ecuador. MG4 and MG5 were close to each other in a branch separated from other MGs. MG4 mainly contained isolates from Ethiopia (85%), while MG5 was an interesting group that contained 20 out of 21 isolates (95.2%) from Mexico and only 1 isolate (4.8%) from Canada. Closely related to each other, MG6, MG7, and MG8 formed a big branch that was most distantly related to the other MGs. These three MGs contained 76.1% of the isolates (51 out of 67) from China and 5.4% of the isolates from Egypt, which comprised 78% of the Egyptian isolates. MG6 was more widely distributed than either MG7 or MG8, as it was detected in all countries except Ethiopia (Table 1). This MG mainly contained isolates from the U.S. (28%), China (26.7%), Italy (13.3%), and Mexico (13.3%). As the second smallest MG and detected in six countries, but not Pakistan, Canada, or Ecuador, MG7 contained 37.5% of the isolates from China, 25% from the U.S., and 18.8% from Italy. Compared to MG6 and MG7, MG8 had a relatively narrow distribution, with 64.1% of the isolates from China, 30.8% from Italy (31%), and only 5.1% (two isolates) from the U.S. MG9 and MG10 were more closely related to each other than to other MGs and formed another big branch. As the largest MG, consisting of 169 isolates or 29.8% of the total isolates and detected in all nine countries, MG9 had the highest percentage of isolates from the U.S. (42%), followed by Mexico (30.2%), Italy (8.3%), and Canada (6.5%). In contrast, MG10 was detected in five countries (the U.S., Italy, Ethiopia, Ecuador, and Mexico), comprising 33.3%, 30.2%, 25.4%, 7.9%, and 3.2% of the isolates from these countries, respectively. 

In addition to the numbers and frequencies of isolates of each MG in each country (Table 1), the proportional presence of MGs in each country can be easily seen in a map (Figure 3). The Chinese collection comprised 37.3% MG8 isolates, followed by 29.9% MG6, 14.9% MG1, 9% MG7, and 9% MG9, but did not have any isolates in MG2 to MG5 or MG10. The small collection from Pakistan had isolates in only three MGs: 50% in MG9 and 25% each in MG3 and MG6. Eight MGs were detected in Italy, including MG1 (50%), MG10 (14.4%), MG9 (10.6%), MG8 (9.1%), MG6 (7.6%), MG3 (5.3%), MG7 (2.3%), and MG2 (0.8%). Nine isolates from Egypt were identified as MG6 (66.7%) and one isolate (11.1%) each in MG1, MG7, and MG9. Six MGs were detected in Ethiopia, including MG4 (52.3%), MG10 (24.6%), MG1 (15.4%), MG9 (4.6%), and MG3 and MG7, each at 1.5%. The Canadian isolates were identified mostly as MG9 (78.6%), with only one isolate (7.1%) each in MG1, MG4, and MG6. Seven MGs were detected in Mexico, including MG9 (56%), MG5 (22%), MG6 (11%), MG1 (6.6%), MG10 (2.2%), and MG2 and MG7, each with only one isolate (1.1%). The Ecuador isolates were identified as MG2 (34.1%); MG9 (18.2%); MG1, MG4, MG6, and MG10 (11.4% each); and MG3 (2.3%). The selected U.S. isolates were identified in all MGs except MG4, with 51.8% in MG9, 15.3% each in MG6 and MG10, 9.5% in MG1, 2.9% in MG7, 2.2% in MG3, 1.5% in MG8, and 0.7% in both MG2 and MG5.

To further detect the relationships between individual isolates from different countries identified in the same MGs, two trees showing individual isolates were constructed using hierarchical cluster analysis, one consisting of all isolates of the closely related MG6, MG7, and MG8 (Figure 4A), and one containing all isolates of MG9 (Figure 4B). These two trees showed that many isolates from the U.S. and Mexico were clustered together, and some were identical. For example, Mexican isolates MX16-249-2 and MX16-244, and U.S. isolates US11-175, US13-272, and US13-226, were identical, as were isolates MX16-237 and US14-215 (Figure 4A). Isolates US10-161 and IT14-28 from Italy, and EC15-43 from Ecuador, were identical; and IT14-5, EG18-9 and EG18-119 from Egypt were identical. US14-309 and MX16-219; PK12-99 and MX16-210; MX16-217-1, US13-019, and US14-176; MX16-211-1 and US14-196; CA13-447 from Canada, US12-153-NG, and US16-170; IT18-13 and US17-061; and more interestingly, 27 Mexican isolates (from MX16-017 to MX16-250) and 5 U.S. isolates (US12-26, US14-11, US14-114, US14-49, and US15-50) were identical (Figure 4B). These results indicate that the migrations of *Pst* urediniospores not only occurred among geographically close countries, such as the U.S. and Mexico, but also between geographically distant countries on different continents. 

### 2.4. Diversity and Heterozygosity in Different MGs and Countries

The diversities of different MGs were determined by the ratio (g/N) of the number of MLGs (g) over the number of isolates (N), the Shannon–Wiener index of genotypic diversity (H), and the Stoddart and Taylor’s genotypic diversity index (G) coupled with respective 95% confidence intervals by bootstrap statistics (Table 2). The ratio of g/N was relatively high (>0.80) in MG1, MG3, MG4, MG6, MG7, and MG10, but relatively low (<0.80) for MG2, MG5, MG8, and MG9, with the highest (1.00) in MG3, the lowest (0.56) in MG2, and a mean of 0.76. The genotypic diversities measured by the H value were relatively high for MG1 (4.60, confidence interval: 4.49–4.72), MG6 (4.06, confidence interval: 3.90–4.22), and MG9 (4.19, confidence interval: 4.00–4.39) compared to the remaining MGs. The confidence intervals of the MG1 values did not overlap with those of the other MGs, showing a significantly higher diversity than any of the other MGs. Similarly, the G value was also relatively high for MG1 (92.24, confidence interval: 84.26–100.21), MG4 (32, confidence interval: 27.83–36.17), MG6 (49.78, confidence interval: 43.02–56.53), MG9 (25.99, confidence interval: 16.18–35.80), and MG10 (31.75, confidence interval: 24.28–39.23) compared to those of the remaining MGs with G values less than 15.00. Both MG1 and MG6 were significantly more diverse than the remaining MGs.

Similarly, the MLG diversities of different countries were also determined (Table 3). The highest genotypic diversity measured by both the H (4.69, confidence interval: 4.58–4.81) and G (96.8, confidence interval: 87.57–106.04) values was found in Italy, followed by the U.S. (H = 4.49, G = 74.78) and Ethiopia (H = 4.06, G = 56.33). Egypt had the lowest genotypic diversity measured by both the H (2.04, confidence interval: 1.63–2.46) and G (7.36, confidence interval: 5.38–9.35).

Private MLGs in each country contributed greatly to the diversity of the global population. The number of private MLGs varied greatly, ranging from 6 in Pakistan to 113 in Italy (Table 3), while the frequencies of private MLGs were all above 80%, except for in Pakistan (75%), with the highest frequency in Ethiopia (98.3%) followed by China (98.0%), Italy (96.6%), and Ecuador (94.7%).

Heterozygosity, which also represents gene diversity, was estimated for each MG and country. The frequencies of homozygous (homokaryotic) and heterozygous (heterokaryotic) alleles across the 14 SSR loci are provided in Appendix A, and the heterozygosity frequencies of the 10 MGs and of the nine countries are provided in Appendix A, respectively. The frequencies of heterozygous alleles of the SSR loci varied greatly, ranging from 5.9% (Pstp029) to 63.5% (CPS02), with a mean of 30.3% across the 567 isolates (Appendix A). The highest mean heterozygous allele frequency was observed in MG9 (57%), followed by MG10 (56%), MG1 (49%), MG3 (35%), MG4 (25%), MG2 (20%), MG7 (18%), and MG8 (15%), with the lowest frequency (14%) in both MG5 and MG6 (Table 2 and Appendix A). For the country-wise comparisons, Canada (49%), the U.S. (48%), Italy (47%), and Mexico (45%) had high mean values of heterozygous allele frequencies; Egypt (18%) and China (21%) had low mean values; and Ethiopia (33%) and Ecuador (27%) were in between (Table 3 and Appendix A). Big ranges of heterozygosity values were observed for every country except Egypt. The results indicate that the *Pst* populations were highly heterozygous, indicating that the two nuclei in most of the isolates were different, and that different alleles between the two nuclei in a single isolate contributed greatly to the population diversity.

### 2.5. Genetic Relationships of Country-Wise Populations

A phylogenetic dendrogram and a scatter plot based on the discriminant analysis of principal component (DAPC) analysis for nine country-wise populations are shown in Figure 5. In the phylogenetic dendrogram (Figure 5A), the Chinese and Egyptian populations were clearly separated from the other country populations by a 100% bootstrap value. The Ethiopian population was also differentiated clearly from the other populations by a 39.8% bootstrap value. The U.S., Canadian, and Mexican populations were clustered together by a 67.5% bootstrap value, and this branch further grouped together with the Ecuadorian population by a 30.8% bootstrap value. Similarly, in the DAPC scatter plot, the isolates from China and Ethiopia were obviously differentiated from each other. Most of the isolates of the U.S., Canada, Mexico, and Ecuador (especially the U.S., Canada, and Mexico) were closely related and distributed in a different coordination from those of the Chinese and Ethiopian isolates (Figure 5B). 

### 2.6. Genetic Variation, Population Differentiation, and Migration

The genetic variations of the *Pst* populations from different countries were analyzed by analysis of molecular variance (AMOVA) (Table 4). The highest variation was among isolates (78%), followed by countries (12%) and within countries (10%). All these levels of variation were highly significant (*P* < 0.001).

The Fixation index (F_ST_) were used to estimate the genetic differentiations among the populations of different countries. The F_ST_ estimations and *P* values of the pairwise comparisons among the nine country populations are shown in Table 5. The differentiations were significant between all country-wise comparations at the *P* = 0.05 level, except between Canada and Mexico, and also at *P* = 0.01, except between Canada and Ecuador, between Canada and Ethiopia, and between Canada and the U.S., in addition to between Canada and Mexico, indicating that these country populations were less differentiated.

A migration network generated based on the Nm values (number of effective migrants) showed the possible migrations between any of the studied countries (Figure 6). Higher migration rates (lower differentiation) were observed among the U.S., Canada, and Mexico populations than between the other countries. Interestingly, the highest migration (1.0) was detected between China and Egypt, and considerable levels of migration (>0.2) were detected between Ethiopia and Canada, Mexico, or Ecuador; between the U.S. and Pakistan or Italy; and between Italy and China or Egypt. These results indicate frequent migrations between different continents.

### 2.7. Reproduction Mode

The standard index of association (rbarD) value was calculated for each of the 10 MGs (Table 2) and each of the nine country populations (Table 3). The rbarD values ranged from 0.006 in MG8 to 0.222 in MG5 (Table 2, Appendix A). The *P* values < 0.05 of five MGs (MG1, MG4, MG5, MG9, and MG10) indicated that their rbarD values were significantly different from the theoretic values for lack of linkage disequilibrium, and thus, the populations were clonal. In contrast, the remaining five MGs (MG2, MG3, MG6, MG7, and MG8) had *P* values > 0.05, indicating that their rbarD values were not significantly different from the theoretical values for lack of linkage disequilibrium, and thus, recombination possibly occurred in these genetic groups. However, the rbarD value of the overall population was 0.116, with a *P* value < 0.001, indicating that the overall population was mostly asexually reproduced.

The rbarD values ranged from 0.087 in Italy to 0.413 in Mexico (Table 3, Appendix A). All country-wise populations had *P* values < 0.001 (Table 3) and were located outside the bell-curve of the simulated distribution (expected from unlinked loci) of a randomly mating population in the *Pst* population (Appendix A), thus, the data reject the hypothesis of no linkage among markers, but support the mostly clonal reproduction of the *Pst* populations based on countries. 

### 2.8. Correlation between the Molecular and Virulence Data

Correlation analysis was conducted with the marker data obtained from this study and the virulence data of a separate study [52] using Mantel tests with 1000 permutations. When all isolates were considered without any grouping, the overall correlation coefficient was 0.235 (*P* < 0.001) between the marker genotypes and virulence phenotypes (Table 6 and Table 7), indicating a low but significant correlation between the two data sets. When the individual MGs were analyzed separately, the correlation coefficients ranged from 0.069 (MG4) to 0.308 (MG2), and significant correlations (*P* < 0.05) were found for MG1, MG2, MG9, and MG10 (Table 6). When the individual country populations were analyzed separately, the highest correlation (0.366) was found in the U.S. population, followed by the Ethiopian population (0.319), and the lowest (0.156) in the Canadian population followed by the Italy population (0.172) and the Chinese population (0.177) (Table 7). The correlations were all significant at *P* = 0.05, except those of the populations from Canada (*P* = 0.203) and Ecuador (*P* = 0.132).

## 3. Discussion

The stripe rust pathogen is distributed in the wheat-growing countries throughout the world, and the fungus evolves and disseminates rapidly from one country to another [9,10,11,13,14,15,21,26,32,33]. The present study successfully tested the hypothesis that *Pst* migrates among countries and continents, which was proposed based on the virulence data of the collections from different countries in a recent study [52]. Our study offers clear evidence supporting *Pst* migration around the global. In addition, our study found great diversities within country-wise populations and significant differences among the country-wise populations and molecular groups. The results of the present study expand our understanding of the mechanisms of the pathogen’s evolution.

In this study, we identified 433 MLGs, including 333 previously undescribed MLGs, from 567 *Pst* isolates collected from nine countries using 14 SSR markers. The 14 markers were adequate to characterize the *Pst* collection, as the marker sufficiency test indicated that 99.5% of the 433 MLGs could be differentiated with only 13 markers (data not shown), which is consistent with our previous study of U.S. collections with the same 14 SSR markers [43]. The 433 MLGs were clustered into 10 MGs, and these MGs had different frequencies and distributions in different countries, and they differed in diversity, heterozygosity, and correlation with virulence phenotypes. Similarly, the country-wise populations also had different levels of diversity and differentiation, and differed in heterozygosity and correlation with virulence phenotypes. However, we detected identical and closely related MLGs in several MGs from different countries. These results indicate the rapid dissemination of specific genetic populations across different continents. Moreover, the analyses of reproduction mode indicate that the overall and country-wise *Pst* populations were generally clonal, but recombination was detected within several MGs. Therefore, this study provides insights for understanding the global diversity, distribution, differentiation, migration, and reproduction of the *Pst* populations.

The geographic distribution and genetic relatedness of the MLGs in different countries were analyzed by the minimum spanning networks by utilizing a stepwise mutation model for the 14 SSR loci. Even though most of the MLGs were private and more closely related within each country, some MLGs from different countries were also interrelated. MLGs from geographically close countries, such as Ecuador, Mexico, and the U.S., and geographically distant countries, such as China, Italy, and the U.S., were both found to be identical or very closely related, indicating migrations or incursions from country to country, which supports the previous reports of the international migration of *Pst* in recent decades [9,13,14,26,32,50,56]. 

The 433 MLGs were clustered into ten MGs, of which some were present in all countries, also indicating *Pst* migration among different countries. Among the ten MGs, MG1, MG6, and MG9 had more isolates and were more diverse and widespread. These MGs consisted of isolates from eight or all nine countries. Although each of the MGs was detected in at least two countries, they appeared to have preferred countries. MG1 and MG3 had the highest numbers of isolates from Italy. MG2 had the highest frequency in Ecuador, MG4 in Ethiopia, MG5 in Mexico, and MG8 in China. MG6 had the highest number of isolates from the U.S., immediately followed by China. MG7 had the highest number of isolates from China followed by the U.S. and Italy. The highest number of isolates of MG9 was from the U.S., followed by Mexico. The U.S. contributed the highest number of isolates to MG10, followed by Italy and Ethiopia. The ten MGs had different levels of relatedness. MG1, MG2, and MG3 were more closely related to each other, and so were MG6, MG7, and MG8. The differences in predominant MGs in different countries may be due to the different foundation populations existing and evolving independently, plus the shared genotypes migrating among countries in recent years. 

In a previous study of only U.S. isolates using the same 14 SSR markers, three major groups (MGs) with 10 subgroups (sub-MGs) were identified [43]. The relationships of the 10 MGs identified in the present study were similar to those of the previously identified sub-MGs. For example, 82% of the U.S. isolates in MG6, MG7, and MG8 in the present study were from the previously identified sub-MGs of MG3, which is a new MG that has only been detected since 2000, and mainly since 2010 in the U.S. [43]. In the present study, MG9 consisted of isolates mostly from the U.S. and Mexico and also contained isolates from all other countries. MG6–MG9 may be related to the aggressive strains widely spreading in the world, as reported in previous studies [11,13,14,33,57]. Some small MGs were genetically distant from other MGs and specific to one or a few countries, such as MG4 in Ethiopia and MG5 in Mexico, indicating the possibility of new emergence or incursions. The other small MGs had close relationships with the large MGs, but were separated into different groups. For example, the second smallest group, MG2, was closely related to the second largest group, MG1. As MG2 was mostly from Ecuador and MG1 widespread in eight countries including Ecuador, MG2 was likely evolved from MG1, and became more adapted to the environment and wheat cultivars in Ecuador. Mostly from China and Italy, MG8 was closely related to MG6, which was detected in all countries except Ethiopia. It is likely that MG8 has recently evolved from MG6, as a larger and more diverse group takes longer to develop than a small and less diverse group. Because *Pst* is capable of long-distance spreading by wind or occasional human activities [9,13], these initially small and distinct groups may undergo future wide distribution. Therefore, even though the genetic groups are unique to certain countries, they need to be continually monitored for possible dispersal to other countries or continents.

Despite its capacity for long-distance migration, the global *Pst* population may be very complex due to a combination of other factors, such as different cropping systems, variable climate conditions, and different geographic features [9]. For example, the highest F_ST_ value of 0.540 was found between China and East Africa using the Stubbs Collections from various countries mainly before the 1990s, indicating a strong population subdivision [1,13]. In our previous study, we found low to moderate levels of differentiation, with the highest F_ST_ value of 0.15 between East Asia and South America in the worldwide collections from 2006–2010 [26]. In the present study, the collections from nine countries were more recent (2010–2018), and the highest F_ST_ value (0.263) was also between China and East Africa (Ethiopia), which was similar to the result of the study by Ali et al. [13]. However, even though most of the pairwise country comparisons for differentiation as measured by F_ST_ values were significant (excluding Egypt and Pakistan, because of small numbers of isolates), the population differentiations were at low to moderate levels, further confirming the international migrations of *Pst*, especially in the more recent years since 2010. We found that, overall, genetic variation mostly existed among isolates (78%), with only 10% within countries and 12% among countries. These values are comparable to the 81% among isolates and 13% among countries in our previous study of the international *Pst* collections of 2006–2010 [26]. This indicates a trend of more rapid globalization in *Pst* populations. With the global spreading of the new aggressive strains, including the sudden appearance of MGs in the U.S. in 2010 and 2011 [43], there have been widespread epidemics of stripe rust, causing huge economic losses in many countries in Central and West Asia, East Africa, and North America in the last decade [5,20,22,23]. Therefore, monitoring *Pst* genotypes is continually needed. 

High levels of genotypic diversity were observed in Italy, China, the U.S., and Ethiopia. These results are consistent with a previous study that also reported high levels of diversity in Asia and the Mediterranean area [26]. This is quite different from the study of Ali et al. (2014), who reported low levels of diversity in the Mediterranean area and North America [13]. In the present study, the highest level of genotypic diversity (4.69 for H and 96.8 for G) was in the Italian population. Italy is part of the Mediterranean region, which was considered a center of origin for *P. striiformis* by Stubbs (1985) [1]. In a recent study analyzing weather conditions for suitability to *Pst* infection, Italy was found among the countries with high risk to stripe rust [58]. The difference could be due to the sizes and regions of *Pst* samples. The North American samples analyzed by Ali et al. (2014) were from southcentral states [13], where *Pst* has a much lower diversity than in the western U.S. [22,23,43,59,60]. In general, *Pst* populations should show high diversity in regions with weather conditions and cropping systems favorable to their survival, infection, and reproduction. All above genetic studies reported that the Asian *Pst* population is relatively highly diverse compared to those of some other regions, and this high diversity may be partially related to possible sexual reproduction in this continent [13,27,28,29,30,31,61]. In 2014, the Himalayan region was proposed as a *Pst* point of origin [13]. In agreement with our previous study, the whole Mediterranean to Asian region could be the main center of origin for the stripe rust fungus. This modified hypothesis is related to the long history of cereal cultivation and stripe rust epidemics in this big geographic region [1,3].

As *Pst* is a dikaryotic fungus, the heterozygosity of isolates makes a major contribution to the overall diversity. Ali et al. (2014) studied the heterozygosity of *Pst* populations in different countries and reported heterozygosity values of 0.28 in China and 0.35 in Pakistan [13]. In the present study, we observed high levels of heterozygosity in Canada (49%), the U.S. (48%), Italy (47%), and Mexico (45%), with the lowest levels in China (21%) and Egypt (18%). The heterozygosity values of China and Pakistan were comparable with those reported by Ali et al. (2014) [13], and the values of Pakistan, Mexico, and the U.S. were consistent with our previous study [43]. However, the heterozygosity value for China in the present study was much lower than that in our previous study (47%) [43]. This difference could be due to the different regions of the collections. In the previous study, the Chinese isolates were from the northwestern region, while in the present study, the samples were from only one province (Yunnan) in the southwestern region. The *Pst* populations in these two regions of China are quite different [30,62]. In addition to the different heterozygosity levels among the country-wise populations, differences in heterozygosity were also detected among the MGs, with the highest heterozygous allele frequencies in MG9 (57%) and MG10 (56%) and the lowest in MG5 and MG6 (both 14%). The heterozygous allele frequencies were significantly correlated to the numbers of isolates of MGs (r = 0.692, *P* = 0.027), which may support the hypothesis that heterozygous isolates may have advantages in pathogen evolution because of the heterosis effect, which may be involved in adaptation to different environments and host cultivars with different resistance genes [26,43]. 

Most previous studies reported the clonal reproduction of *Pst* populations in many parts of the world [26,29,36,37,38,43,57,63]. Some studies on population genetics have suggested the sexual recombination of *Pst* in China and the Himalayan region [13,30,31,61], and the natural infection of *Pst* in barberry for sexual reproduction was reported at low frequencies in China [25,27,28,29]. In the present study, clonal reproduction was indicated for each individual country and also for the overall population. However, when the isolates in the 10 MGs were analyzed separately for the standardized index of association (rbarD), 5 MGs (MG2, MG3, MG6, MG7, and MG8) were found to have recombination at the *P* = 0.01 level, and at the *P* = 0.05 level, 3 MGs (MG2, MG3, and MG8) were still considered to have possible recombination. Based on the lowest rbarD value (0.006) and the highest *P* value (0.309), MG8 is most likely to be produced by sexual recombination. In fact, most MG8 isolates were from China and Italy. China has been shown to have sexual reproduction [25,27,28,29], and Italy has been reported to be at a high potential risk of *Pst* infection on barberry [58]. Thus, this study provides evidence for *Pst* sexual recombination in some countries, especially China. Not all possibilities of recombination are through sexual reproduction, and some of them may be results of somatic recombination. Somatic recombination has been demonstrated for *Pst* under controlled conditions [44], and has been reported to produce virulent races in *Puccinia graminis* f. sp. *tritici*, the wheat stem rust pathogen [64,65]. Nevertheless, among all possible mechanisms, mutation is the most important for *Pst* evolution, which is illustrated by the minimum span of the network for stepwise mutations to produce the majority of MLGs in the present study (Figure 1), as is consistent with our previous studies [26,43].

The overall correlation coefficient value (0.235) between the molecular genotype data and the previously published virulence data [51,52] was low, but significant. This value is lower than the 0.41 value reported for the international collections from 2006 to 2010 [26]. The difference could be due to the collections being from different countries in the two studies, as different country populations had different coefficient values in the present study. The U.S. and Ethiopian populations had the highest correlations, while the Ecuadorian population had the lowest correlation. Similarly, the different MGs had different coefficient values, with MG2 having the highest (0.308) and MG4 the lowest (0.069). The correlations in five MGs were significant, but they were insignificant in the other five MGs. The differences in correlation may indicate different deployments of specific resistance genes in different countries, and some MGs have been under stronger wheat cultivar selection pressure than others. Because the overall correlation is not very high, neither virulence nor molecular tests can provide a complete picture of the pathogen population. Thus, it is important to continually characterize *Pst* populations using both virulence testing and molecular markers. 

## 4. Materials and Methods

### 4.1. Sample Collection and Urediniospore Multiplication

A total of 567 *Pst* isolates from 9 countries were used in this study, including 132 U.S. isolates selected to represent different races and molecular groups during the same period of collection in the other countries for the purpose of genotype comparison (Table 1, Appendix A). As the quantity of urediniospores for each isolate stored in liquid nitrogen was limited, urediniospores were multiplied on seedlings of wheat cultivar “Nugaines”, which is susceptible at the seedling stage to all *Pst* races identified thus far in the U.S. [22,23]. After heat shock at 50 °C for 2 min, urediniospores sealed in small foil bags were transferred with a fine brush onto two-leaf-stage seedlings of Nugaines. The inoculated plants were incubated in a dew chamber at 10 °C for 24 h without light, and grown in a growth chamber with a diurnal temperature cycle gradually changing from 4 °C at 2:00 a.m. to 20 °C at 2:00 p.m., with 8 h dark in the low temperature range and 16 h light in the high temperature range [59,60]. To prevent cross contamination, plants inoculated with different isolates in different pots were separated with plastic booths. Urediniospores were vacuum collected with custom-made glass collectors.

### 4.2. Genomic DNA Extraction

Total genomic DNA was extracted from dried urediniospores following a universal and rapid salt-extraction method for high-quality genomic DNA [66] with modifications for *Pst* [67]. A mixture of ~20 mg spores and 200 mg sand in a 1.1 mL tube was ground by vortexing for 2 min and added to 500 µL 2× cetyltrimethylammonium bromide (CTAB) buffer (1.4 M NaCl, 100 mM Tris-HCl pH 8.0, and prewarm to 65 °C) (ThermoFishher Scientific, Waltham, MA, USA). After mixing, the tube was incubated in a water bath at 65 °C for 60 min with gentle inverting every 10 min, and mixed with 400 μL chloroform: iso-pentyl alcohol (24:1) (ThermoFishher Scientific, Waltham, MA, USA), then centrifuged for 15 min at 4000 rpm at 4 °C in an Allegra 25R centrifuge (Beckman Coulter Inc., Brea, CA, USA). The 500 μL supernatant was transferred to a new tube and mixed with 500 μL isopropanol; the mixture was centrifuged for 15 min and the DNA pellet was air-dried and re-suspended in 100 μL TE buffer (containing 20 μg mL^−1^ Rnase A). The tube was incubated at 37 °C for 2 h to completely dissolve the DNA pellet. The concentration of the DNA stock solution was determined using a ND-1000 spectrophotometer (Bio-Rad, Hercules, CA, USA), and the quality was checked in a 0.8% agarose gel. A work solution of 5 ng µL^−1^ was made from the stock solution by adding sterile deionized water for use as a DNA template in the polymerase chain reaction (PCR). 

### 4.3. SSR Markers and PCR Amplification

Fourteen pairs of SSR primers were selected based on their co-dominant polymorphisms among *Pst* isolates shown in previous studies [26,35,36,39]. They were CPS02, CPS04, CPS08, and CPS13 [68]; PstP001, PstP002, PstP003, PstP005, PstP006, and PstP029 [35]; and RJ18, RJ20, RJ21, and RJ8N [69,70]. The primers were synthesized by Sigma Life Science (St. Louis, MO, USA). The sequences, annealing temperatures, and amplied fragments of the primers are provided in Appendix A. To use fluorescence for detecting PCR products, an M13 tag (5′-CACGACGTTGTAAAACGAC) was added to the 5′ end of each forward primer [48]. For each SSR marker, the forward primer was labeled with black, green, or blue florescent dye, with the red fluorescent dye (Sigma Life Science) for the size marker. SSR loci with the same or similar allele sizes were labeled with different florescent dyes to achieve the maximum possible number of loci per run in the sequencer [71]. 

Amplification of PCR was performed in a Bio-Rad iCycler (Bio-Rad, Hercules, CA, USA) following the protocol described in previous studies [35,36]. Each reaction (12 µL) contained 1.2 µL of 10× reaction buffer with 15 mM MgCl_2_ (Sigma Life Science), 0.96 µL 2.5 mM dNTP (Sigma Life Science), 0.12 µL of 5 µM forward primer, 0.6 µL 5 µM reverse primer, 0.24 µL of 5 µM M13 universal primer, 0.2 µL 5 U/µL *Taq* polymerase (Sigma Life Science), 4 µL DNA (total 20 ng), and 4.68 µL sterile ddH_2_O. The amplification cycles and conditions were 94 °C for 5 min for initial denaturation; 42 cycles of 94 °C for 30 s, 45 to 54 °C for 30 s depending on primers, and 72 °C for 45 s; and 7 min of final extension at 72 °C. The sizes of the PCR products were estimated using capillary electrophoresis on an ABI3730 Genotyper (Applied Biosystems, Foster City, CA, USA). The internal molecular weight standard for ABI3730 was Genescan 445-LIZ (Applied Biosystems). Allele sizes in base pairs were scored and analyzed using the software GeneMarker V2.2 (Softgenetics, State College, PA, USA). 

### 4.4. Analyses of Multilocus Genotypes

If isolates in the present study had identical alleles across all 14 SSR loci, they were assigned to the same MLG. The MLGs were named in continuation with the previous MLGs identified from the U.S. isolates of 2010 to 2017 [43]. Since *Pst* is a dikaryotic fungus at the uredinial stage, each isolate was scored for two alleles to determine homozygous or heterozygous for each SSR locus using GeneMarker V2.2 (https://genemarker.software.informer.com/2.2/ (accessed on 20 August 2021)). The data file generated by GeneMarker was converted to a ‘csv’ file to be analyzed with the R package. The sufficiency of markers to describe the population structure was assessed through the detection of multilocus genetypes (MLGs) plotted against the number of loci, and a graph was generated using the ‘*poppr’* package in the R 4.0.3 program (https://r-for-windows.updatestar.com/ (accessed on 20 August 2021)) [72]. The genetic relationships among individual MLGs were analyzed based on the Bruvo’s distance utilizing a stepwise mutation model for microsatellite loci [73,74,75] and visualized by a minimum spanning network in ‘*poppr*’. 

### 4.5. Identification of Molecular Groups

To identify putative clusters of genetically related isolates in the present study (567 isolates), hierarchical clustering analysis based on the 14 SSR markers (CPS02, CPS04, CPS08, CPS13, RJ18, RJ20, RJ21, RJ8N, Pstp001, PstP002, PstP003, Pstp005, PstP006, and PstP029) was conducted using the dissimilarity values and the “ward.D2” method with the “hclust” function in the R stats 4.0.3 program [76]. The parameters for hierarchical cluster analysis were the same as previously described [43]. 

### 4.6. Population Diversity and Heterozygosity

Genotypic diversity was determined by estimating both genotypic richness (the number of observed MLGs) and evenness (the distribution of genotype abundance). Stoddart and Taylor’s index and Shannon’s diversity index for MLG diversity were calculated as G = 1/∑P_i_^2^ and H = ∑P_i_lnP_i_, respectively, where P_i_ is the observed frequency of the ith MLG in a population [53,54,77]. As *Pst* is a dikaryotic fungus at the uredinial stage and DNA was extracted from urediniospores, each isolate was scored for homozygous or heterozygous at each SSR locus using the software GeneMarkerV1.5 (https://genemarker.software.informer.com/download/ (accessed on 20 August 2021)). Heterozygosity in percentage was calculated for each isolate based on the number of loci with different alleles across the 14 SSR markers analyzed using the software GenAlEx 6.503 (https://biology-assets.anu.edu.au/GenAlEx/Welcome.html (accessed on 20 August 2021)) [78]. Mean heterozygosity was calculated for each country and MG.

### 4.7. Population Variation, Differention, and Phylogenetic Relationships

Analysis of molecular variance (AMOVA), which allows the hierarchical partitioning of genetic variation within and among countries [79], was conducted by defining the *Pst* isolates based on countries using the function “poppr.amova” in the *poppr* R program version 4.0.3 [72,80]. Population differentiation measured by the fixation index (F_ST_) among the nine countries, except for Pakistan and Egypt which had isolate numbers less than ten, were analyzed using the software GenAlEx 6.503 [78]. A migration network based on the number of effective migrants (Nm, where N is the effective population size of each population and m is the migration rate between populations) was also generated to visualize the gene flow patterns among the nine countries using the “diversity” package in R. To further investigate the genetic relationships at the country level, a phylogenetic tree was generated based on Edward’s genetic distance for the nine countries in bootstrap analysis with 1000 replicates using function “aboot” in the “poppr” program. Similarly, to assess how the nine country-wise populations differed from each other, the discriminant analysis of principal components (DAPC), a no model-based method developed and implemented in the “*adegenet”* R package, was also performed to generate a scatter plot. The DAPC was carried out for all marker loci, and an α-score optimization was used to determine the number of principal components to retain [81,82]. 

### 4.8. Reproduction Mode

The mode of reproduction was tested by adjusted/standard index of association (rbarD) [55]. This test is useful to determine if a populations is clonal (rbarD = 1, where significant disequilibrium is expected due to linkage among loci) or sexual (rbarD = 0, where linkage among loci is not expected). The null hypothesis is that alleles observed at different loci are not linked if the population is sexually reproduced, and they recombine freely into new genotypes during the process of sexual reproduction. In molecular ecology, we typically use the index of association or related indices to test this phenomenon. In this study, rbarD was calculated using the “*poppr*” program, and the null hypothesis rbarD = 0 was tested with 1000 permutations [72]. If an observed rbarD value is located outside the distribution of the randomized dataset at *P* < 0.001, then the samples are likely from a clonal population [72].

### 4.9. Determination of Correlation between the Molecular and Virulence Data

To determine the correlations between the molecular data obtained in the present study with the virulence data in a previous study from our group [52], a distance matrix was generated for each country population with each of the two data sets using GenAlEx 6.5 [78]. The regression analysis was used to obtain the correlation coefficient based on the distance matrix generated from the two datasets, and the significance of correlation was determined using *P* = 0.05. 

## 5. Conclusions

In this study, we identified 433 MLGs from the *Pst* collections of nine countries on five continents in recent years, and clustered them into 10 MGs. These MGs differed greatly in frequency, distribution, genetic diversity, heterozygosity, and correlation coefficient between the molecular genotypes and virulence phenotypes. Similarly, we also detected differences in these aspects among the nine country-wise populations. Although significant differentiations were observed between all possible pairwise country populations at *P* = 0.05, except between Canada and Mexico, the greater the distance between two countries, the greater the differentiation between the two countries’ populations. The presence of the same MGs and the detection of identical or closely related MLGs in different countries indicated frequent migrations of the pathogen among countries and continents. The analysis of the standardized index of association indicated possible sexual reproduction involved in some of the MGs, although the overall population or individual country populations were found to be clonal. The relatively low correlation coefficient values obtained for the overall population, individual country populations, and MGs suggest the need for characterization of *Pst* populations using both virulence phenotyping and molecular genotyping. This study provides insights for understanding the diversity, distribution, and evolution of *Pst* populations on a global scale, and the information is useful for continual monitoring of the pathogen populations. The detection of global migrations of *Pst* genotypes in the present study and virulent races in the previous studies emphasizes the importance of considering the global pathogen populations when breeding wheat cultivars with broad, durable resistance, for the sustainable control of stripe rust.

## Figures and Tables

**Figure 1 ijms-22-09457-f001:**
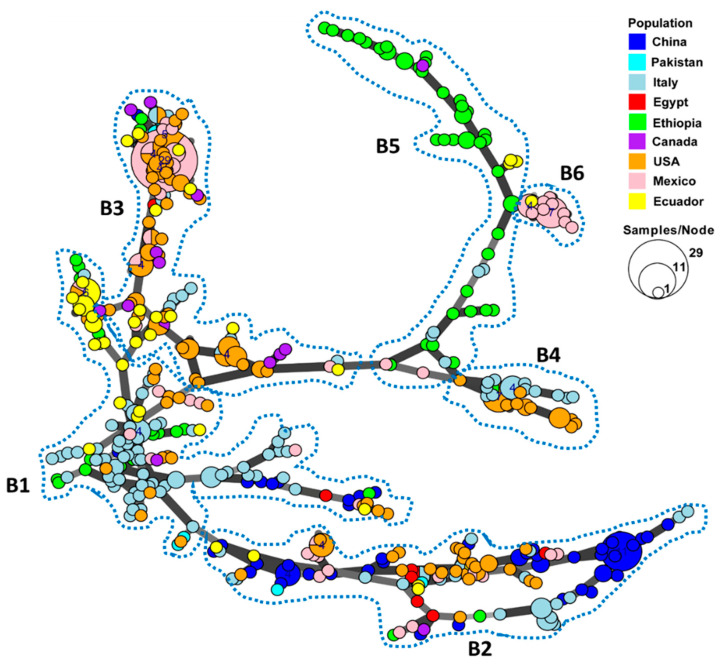
Minimum spanning network showing relationships among multi-locus genotypes (MLGs) of *Puccinia striiformis* f. sp. *tritici* identified from nine countries. Each node represents a different MLG. The size of a node indicates the number of isolates in the MLG. Node colors represent countries from which the isolates were collected. Six major branches are marked by the broken close lines. B1–B6, branch 1–branch 6.

**Figure 2 ijms-22-09457-f002:**
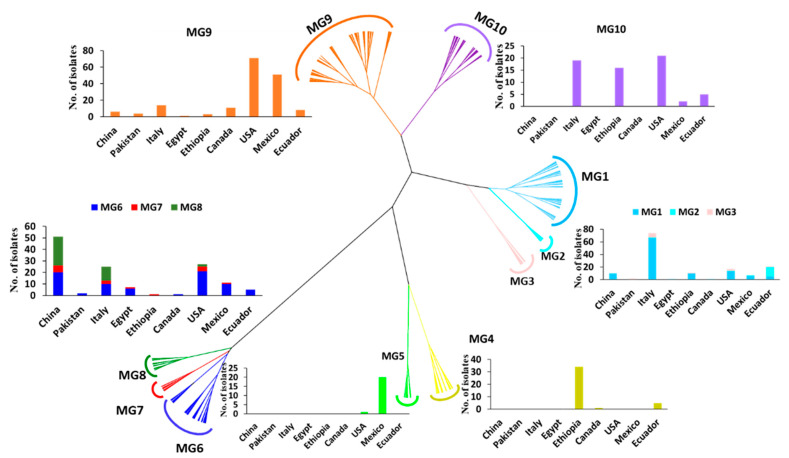
Dendrogram of *Puccinia striiformis* f. sp. *tritici* populations from nine countries constructed based on dissimilarities assessed with 14 simple sequence repeat (SSR) markers using the hierarchical cluster analysis, showing ten molecular groups (MGs) and the histograms with isolate numbers from different countries within each MG.

**Figure 3 ijms-22-09457-f003:**
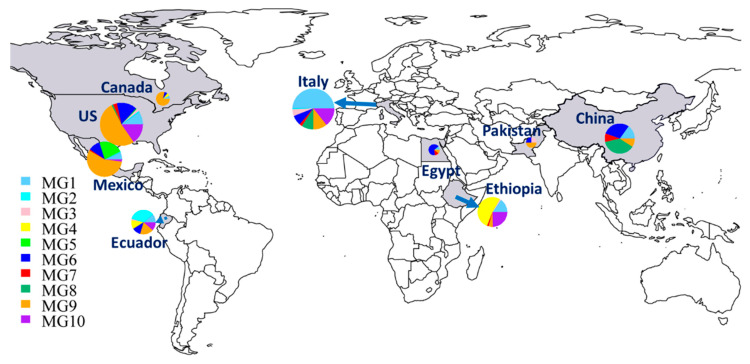
Distributions of molecular groups (MGs) of *Puccinia striiformis* f. sp. *tritici* in nine countries. The sizes of the circles and fans indicate the relative numbers of isolates in the countries and MGs, respectively.

**Figure 4 ijms-22-09457-f004:**
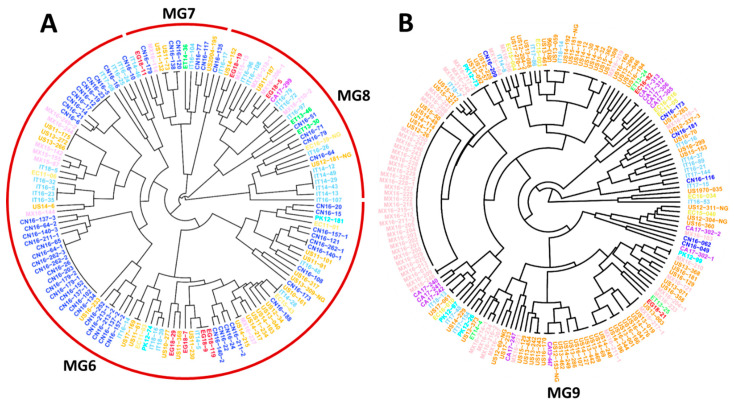
Dendrograms of major molecular groups (MGs) of *Puccinia striiformis* f. sp. *tritici* showing genetically related isolates from different countries. (**A**) Dendrogram of isolates in MG6, MG7, and MG8. (**B**) Dendrogram of isolates in MG9. These dendrograms were constructed based on dissimilarities assessed with 14 simple sequence repeat markers (SSR) using hierarchical cluster analysis.

**Figure 5 ijms-22-09457-f005:**
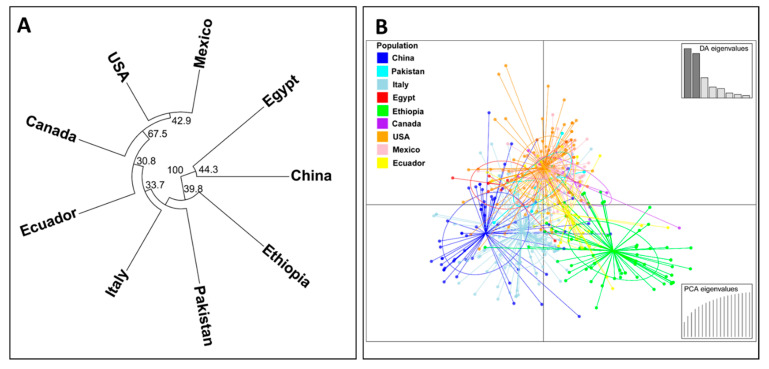
A phylogenetic tree showing the relationships of *Puccinia striiformis* f. sp. *tritici* (*Pst*) collections from nine countries based on the Edward’s genetic distance (**A**) and a scatter plot based on the discriminant analysis of principal component (DAPC) analysis of all *Pst* isolates across the nine countries (**B**).

**Figure 6 ijms-22-09457-f006:**
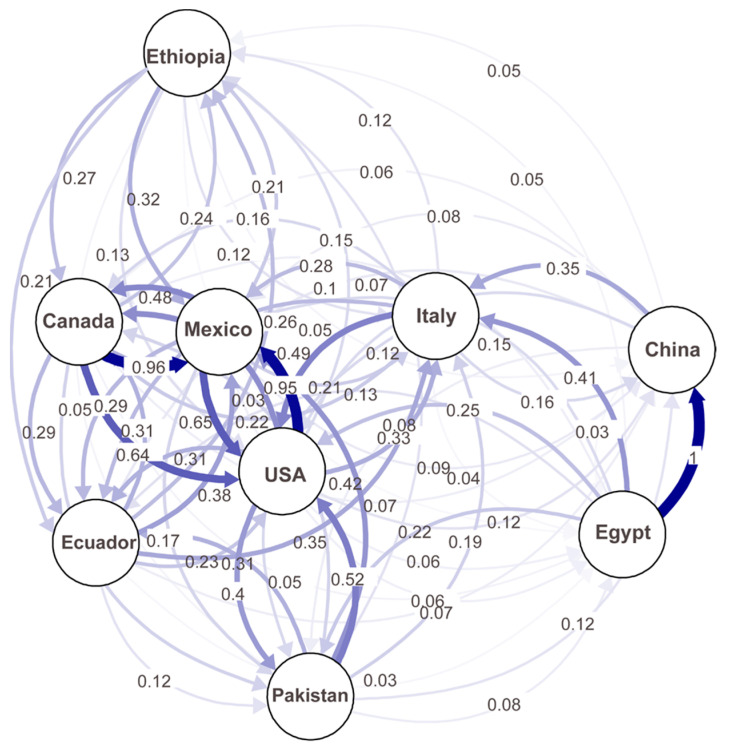
A migration network showing the gene flow patterns among nine country populations of *Puccinia striiformis* f. sp. *tritici*. Gene flow strengths are indicated by the line thicknesses.

**Table 1 ijms-22-09457-t001:** The numbers and frequencies, sampling years, and moleular groups (MG) of the *Puccinia striiformis* f. sp. *tritici* isolates from nine countries in 2010–2018.

	Isolates		Number of Isolates (Frequency, %) in Molecular Groups (MG)									
Country	Number	Year	MG1	MG2	MG3	MG4	MG5	MG6	MG7	MG8	MG9	MG10
China	67	2016	10 (14.9)	0	0	0	0	20 (29.9)	6 (9.0)	25 (37.3)	6 (9.0)	0
Pakistan	8	2012	0	0	2 (25.0)	0	0	2 (25.0)	0	0	4 (50.0)	0
Italy	132	2014, 2016, 2017, 2018	66 (50.0)	1 (0.8)	7 (5.3)	0	0	10 (7.6)	3 (2.3)	12 (9.1)	14 (10.6)	19 (14.4)
Egypt	9	2018	1 (11.1)	0	0	0	0	6 (66.7)	1 (11.1)	0	1 (11.1)	0
Ethiopia	65	2013, 2014	10 (15.4)	0	1 (1.5)	34 (52.3)	0	0	1 (1.5)	0	3 (4.6)	16 (24.6)
Canada	14	2013, 2017	1 (7.1)	0	0	1 (7.1)	0	1 (7.1)	0	0	11 (78.6)	0
Mexico	91	2015, 2016	6 (6.6)	1 (1.1)	0	0	20 (22.0)	10 (11.0)	1 (1.1)	0	51 (56.0)	2 (2.2)
Ecuador	44	2011, 2015, 2016	5 (11.4)	15 (34.1)	1 (2.3)	5 (11.4)	0	5 (11.4)	0	0	8 (18.2)	5 (11.4)
USA	137	2010-2017	13 (9.5)	1 (0.7)	3 (2.2)	0	1 (0.7)	21 (15.3)	4 (2.9)	2 (1.5)	71 (51.8)	21 (15.3)
Total	567	2010-2018	112 (19.8)	18 (3.2)	14 (2.5)	40 (7.1)	21 (3.7)	75 (13.2)	16 (2.8)	39 (6.9)	169 (29.8)	63 (11.1)

**Table 2 ijms-22-09457-t002:** Numbers of *Puccinia striiformis* f. sp. *tritici* isolates, numbers of multi-locus genotypes (MLGs), g/N values, heterozygosity, Stoddart and Taylor’s MLG diversity (G) and confidence interval (CI), and Shannon–Wiener index of MLG diversity (H) over 14 SSR loci, as well as standardized index of association (rbarD) and its probability (*P*) for indication of reproduction mode in ten molecular groups (MGs).

MG	No. of Isolates (N)	No. of MLGs (g)	g/N	Genotypic Diversity (H) (CI) ^a^	Genotypic Diversity (G) (CI) ^b^	Heterozygosity (%) ^c^	Standardized Index of Association (rbarD) ^d^	*P* Value (rbarD)
MG1	112	104	0.93	4.60 (4.49, 4.72)	92.24 (84.26, 100.21)	49 (11–91)	0.063	<0.001
MG2	18	10	0.56	2.03 (1.63, 2.44)	5.79 (3.42, 8.15)	20 (0–94)	0.016	0.286
MG3	14	14	1.00	2.64 (2.34, 2.94)	14 (11.52, 16.48)	35 (0–80)	0.039	0.066
MG4	40	35	0.88	3.52 (3.32, 3.71)	32 (27.83, 36.17)	25 (0–90)	0.080	<0.001
MG5	21	12	0.57	2.13 (1.72, 2.55)	5.88 (3.29, 8.47)	14 (0–91)	0.222	<0.001
MG6	75	64	0.85	4.06 (3.90, 4.22)	49.78 (43.02, 56.53)	14 (3–50)	0.016	0.032
MG7	16	14	0.88	2.60 (2.29, 2.91)	12.8 (10.13, 15.47)	18 (6–60)	0.032	0.033
MG8	39	23	0.59	2.75 (2.40, 3.09)	9.22 (5.19, 13.25)	15 (0–74)	0.006	0.309
MG9	169	105	0.62	4.19 (4.00, 4.39)	25.99 (16.18, 35.80)	57 (2–91)	0.091	<0.001
MG10	63	51	0.81	3.76 (3.55, 3.98)	31.75 (24.28, 39.23)	56 (3–90)	0.107	<0.001
Total/mean	567	432	0.76	5.82 (5.74–5.91)	173.50 (141.82, 205.17)	30 (0–94)	0.116	<0.001

^a^ Shannon–Wiener Index of MLG diversity and confidence interval [53] calculated using the “poppr” package in R program 4.3. ^b^ Stoddart and Taylor’s Index of MLG diversity and confidence interval [54] calculated using the “poppr” package in R program 4.3. ^c^ Mean of heterozygous loci of 14 simple sequence repeats; values in the parentheses are ranges. ^d^ Standardized index of association, rbarD. *P* < 0.001 indicates significant disequilibrium due to linkage among loci, indicating populations are clonal [55].

**Table 3 ijms-22-09457-t003:** Numbers of *Puccinia striiformis* f. sp. *tritici* isolates, numbers of multi-locus genotypes (MLGs), g/N values, heterozygosity, Stoddart and Taylor’s MLG diversity (G) and confidence interval (CI), and Shannon–Wiener index of MLG diversity (H) over 14 SSR loci, as well as standardized index of association (rbarD) and its probability (*P*) for indication of reproduction mode in the collections from nine countries.

Country	No. of Isolates (N)	No. of MLGs (g)	g/N	No. and Freq. (%) of Private MLGs ^a^	Genotypic Diversity (H) (CI) ^b^	Genotypic Diversity (G) (CI) ^c^	Heterozygosity (%) ^d^	Standardized Index of Association (rbarD) ^e^	*P* Value (rbarD)
China	67	51	0.76	50 (98.0)	3.67 (3.42, 3.92)	23.02 (15.22, 30.82)	21 (2–75)	0.115	<0.001
Pakistan	8	8	1.00	6 (75.0)	2.08 (1.69, 2.47)	8 (6.09, 9.91)	38 (0–75)	0.177	<0.001
Italy	132	117	0.89	113 (96.6)	4.69 (4.58, 4.81)	96.8 (87.57, 106.04)	47 (10–79)	0.087	<0.001
Egypt	9	8	0.89	7 (87.5)	2.04 (1.63, 2.46)	7.36 (5.38, 9.35)	18 (0–33)	0.183	<0.001
Ethiopia	65	60	0.92	59 (98.3)	4.06 (3.92, 4.22)	56.33 (50.76, 61.90)	33 (0–89)	0.092	<0.001
Canada	14	14	1.00	13 (92.9)	2.64 (2.35, 2.93)	14 (11.57, 16.43)	49 (7–86)	0.093	<0.001
USA	137	102	0.74	88 (86.2)	4.49 (4.37, 4.62)	74.78 (66.10, 83.45)	48 (9–76)	0.160	<0.001
Mexico	91	53	0.58	44 (83.0)	3.30 (3.00, 3.61)	10.29 (5.63, 14.95)	45 (2–82)	0.413	<0.001
Ecuador	44	38	0.86	36 (94.7)	3.55 (3.34, 3.77)	30.25 (25.04, 35.46)	27 (7–67)	0.088	<0.001
Total	567	432	0.76	416 (96.3)	5.82 (5.74, 5.91)	173.50 (141.82, 205.17)	36 (0–89)	0.116	<0.001

^a^ Private MLGs were those found in one or more isolates in the individual country but not found in any other country. ^b^ Shannon–Wiener Index of MLG diversity and confidence interval [53] calculated using the “poppr” package in R program 4.3. ^c^ Stoddart and Taylor’s Index of MLG diversity and confidence interval [54] calculated using the “poppr” package in R program 4.3. ^d^ Mean of heterozygous loci of 14 simple sequence repeats; values in the parentheses are ranges. ^e^ Standardized index of association, rbarD. *P* < 0.001 indicates significant disequilibrium due to linkage among loci, indicating populations are clonal [55].

**Table 4 ijms-22-09457-t004:** Analysis of molecular variance for partitioning the variation in *Puccinia striiformis* f. sp. *tritici* isolates from nine countries.

Source of Variation	Df ^a^	Sum of Squares	Mean Squares	Estimated Variance	Variation (%)	*P* Value ^b^
Among countries	8	427.84	53.48	0.43	12	<0.001
Within countries	558	1880.26	3.37	0.33	10	<0.001
Among isolates	567	1541.50	2.72	2.72	78	<0.001
Total	1133	3849.59		3.47	100	

^a^ Df = degrees of freedom. ^b^
*P* values were based on 1000 permutations.

**Table 5 ijms-22-09457-t005:** Pairwise comparison of populations of *Puccinia striiformis* f. sp. *tritici* from seven countries based on the fixation index (F_ST_) values ^a^.

Country	Canada	China	Ecuador	Ethiopia	Italy	Mexico	USA
Canada		0.001	0.002	0.002	0.001	0.063	0.015
China	0.184		0.001	0.001	0.001	0.001	0.001
Ecuador	0.051	0.169		0.001	0.001	0.001	0.001
Ethiopia	0.057	0.263	0.062		0.001	0.001	0.001
Italy	0.066	0.066	0.048	0.115		0.001	0.001
Mexico	0.023	0.144	0.048	0.055	0.055		0.001
USA	0.027	0.103	0.056	0.099	0.035	0.020	

^a^ The values below the blank diagonal are F_ST_ values, and those above the diagonal line are *P* values based on 1000 permutations. The Egyptian and Pakistani collections were excluded from the analysis because their isolate numbers were <10.

**Table 6 ijms-22-09457-t006:** Correlation coefficients between the molecular data (SSR) and virulence data within different molecular groups (MG) of *Puccinia striiformis* f. sp. *tritici*.

MG	Correlation Coefficient	*P* ^a^
MG1	0.282	<0.001
MG2	0.308	0.028
MG3	0.153	0.160
MG4	0.069	0.281
MG5	0.259	0.104
MG6	0.093	0.087
MG7	0.087	0.266
MG8	0.075	0.205
MG9	0.293	<0.001
MG10	0.235	<0.001
Overall	0.235	<0.001

^a^ Probability (*P*) values based on 1000 permutations.

**Table 7 ijms-22-09457-t007:** Correlation coefficients between the molecular data (SSR) and virulence data of *Puccinia striiformis* f. sp. *tritici* collections from different countries.

Country ^a^	Correlation Coefficient	*P* ^b^
Canada	0.156	0.203
China	0.177	0.009
Ecuador	0.123	0.132
Ethiopia	0.319	<0.001
Italy	0.172	<0.001
Mexico	0.203	0.005
USA	0.366	<0.001
Overall	0.235	<0.001

^a^ Pakistan and Egypt were excluded from this analysis because their isolate numbers were <10. ^b^ Probability values based on 1000 permutations.

## Data Availability

The data presented in this study are available in the Appendix A.

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
