# Peer review of "Molecular Characterization of Wheat Stripe Rust Pathogen (Puccinia striiformis f. sp. tritici) Collections from Nine Countries"

_ijms, 2021, doi:10.3390/ijms22179457_

Round 1

Reviewer 1 Report

The manuscript is well written and provides important information on the molecular characterization of wheat strip rust collections from nine countries.

I only have the following comments for the improvement of the manuscript:

The title is long, consider shortening it.

Are the 14 codominant SSR markers enough to accurate genotype 567 isolates?

In the abstract, please state the importance of your research results for the scientists and the farmers. The statement that “the study provides information which should be useful to develop strategies for managing stripe rust…” is very general and does not say much. What can be done based on your results?

What is the innovation of the study?

In conclusions, please state the practicality of your research study, meaning what are the practical applications and innovations stemming from your study.

Reviewer 2 Report

The work is well written and results are well discussed. Experimental procedure adopted is very valid. The figures are clear and well represent the results obtained. The work is excellent and from my point of view it can be accepted in th present form.

Author Response

Thanks for taking your time in reviewing our manuscript and the positive evaluation of this work.

Reviewer 3 Report

Major Revision.
I reviewed the manuscript titled, Molecular Characterization of
Wheat Stripe Rust Pathogen (Puccinia striiformis f. sp. tritici)
Collections from Nine Countries Revealed Different Levels of
Diversity, Migration, and Possible Sexual Recombination and my comments are as below.

The abstract must include data regarding the critical finds by the authors in terms of data of important findings.
The introduction must have a clear hypothesis and significantly develop the second paragraph of this manuscript.
Overall there is the repetition of the information which could be avoided.
 Check figure ligands; they are carelessly written.
L141-145 can be divided into understandable sentences.
Discussion should include more information and references related to the relevant and related works.
Restructure and carefully edit the conclusion section, especially at the L292-297 redundancy. 

Reviewer 4 Report

In the current context of global food security issues, the manuscript is well documented, containing sufficient data on the analysis of samples collected from 9 countries over 8 years.

However, in the logic of writing a manuscript, it is mandatory to place the Materials and methods section before the Results section. Re-order the sections and correct the numbering of all subsections, Tables, Figures and bibliographic sources accordingly.

The experimental results are interpreted graphically and systematically analyzed, which guarantees that the manuscript can serve as an interesting reference study for other authors.

The title of manuscript is quite long and it sounds more like an overall conclusion of the study. I would advise the authors to use just “Molecular Characterization of Wheat Stripe Rust Pathogen (Puccinia striiformis f. sp. tritici) Collections from Nine Countries”. The existence of “Different Levels of Diversity, Migration, and Possible Sexual Recombination” can be specified in the Abstract.

Keywords – this section should not repeat words from the title.

Check the manuscript for missing commas.

Line 74 – instead of “more recent populations are needed to be characterized to determine”, write “more recent populations must be characterized to determine…”.

Line 122 – Correct to “All the MLGs could be separated into six branches which were labelled by the broken line”.

Line 125 – Figure

Lines 170, 228, 456 – use the hyphen.

Line 309-310 – “The highest variation was found among isolates (78%), followed by countries (12%), and within countries (10%) respectively.”

Line 444 – “In our previous study, we found low to moderate levels”

Line 461 – “especially the invasive groups”.

Line 464-465 – “This is quite different from the study of 464 Ali et al. (2014), who reported low levels”.

Line 471 – “The North American samples analyzed by Ali et al. (2014) were from south-central states, [13], where Pst has much low diversities than in the western U.S.”.

Line 483 – “Ali et al. (2014) studied the heterozygosity”.

Lines 485-486 – “In the present study, we observed high levels of heterozygosity in Canada (49%), the U.S. (48%), Italy (47%), and Mexico (45%) respectively, with the lowest levels in China (21%) and Egypt (18%).”

Line 503 – “Some studies on the population genetics suggested sexual”.

Line 522 – Figure 1.

Lines 529-530 – “The U.S. and Ethiopian populations had the highest correlation, while the Ecuadorian population had the lowest correlation.”

Lines 536-537 – “Thus, it is important the continuous monitoring of Pst populations using both virulence testing and molecular characterization”.

In Conclusions section, use the Past Tense.

Round 2

Reviewer 3 Report

Authors have significantly improved the manuscript, therefore it can be accepted after a careful check by the editors for English and other important aspects.